# Molecular Cytogenomic Characterization of the Murine Breast Cancer Cell Lines C-127I, EMT6/P and TA3 Hauschka

**DOI:** 10.3390/ijms21134716

**Published:** 2020-07-01

**Authors:** Shaymaa Azawi, Thomas Liehr, Martina Rincic, Mattia Manferrari

**Affiliations:** 1Jena University Hospital, Friedrich Schiller University, Institute of Human Genetics, Am Klinikum 1, D-07747 Jena, Germany; shayma.alazawi@yahoo.com (S.A.); mattiamanferrari@gmail.com (M.M.); 2Croatian Institute for Brain Research, School of Medicine University of Zagreb, Salata 12, 10000 Zagreb, Croatia; mrincic@hiim.hr

**Keywords:** breast cancer, murine cell line, C-127I, EMT6/P, TA3 Hauschka, murine multicolor banding (mcb), array comparative genomic hybridization (aCGH), estrogen receptor (ER), progesterone receptor (PR), human epidermal growth factor receptor-2 (HER-2) receptor

## Abstract

Background: To test and introduce effective and less toxic breast cancer (BC) treatment strategies, animal models, including murine BC cell lines, are considered as perfect platforms. Strikingly, the knowledge on the genetic background of applied BC cell lines is often sparse though urgently necessary for their targeted and really justified application. Methods: In this study, we performed the first molecular cytogenetic characterization for three murine BC cell lines C-127I, EMT6/P and TA3 Hauschka. Besides fluorescence in situ hybridization-banding, array comparative genomic hybridization was also applied. Thus, overall, an in silico translation for the detected imbalances and chromosomal break events in the murine cell lines to the corresponding homologous imbalances in humans could be provided. The latter enabled a comparison of the murine cell line with human BC cytogenomics. Results: All three BC cell lines showed a rearranged karyotype at different stages of complexity, which can be interpreted carefully as reflectance of more or less advanced tumor stages. Conclusions: Accordingly, the C-127I cell line would represent the late stage BC while the cell lines EMT6/P and TA3 Hauschka would be models for the premalignant or early BC stage and an early or benign BC, respectively. With this cytogenomic information provided, these cell lines now can be applied really adequately in future research studies.

## 1. Introduction

Breast cancer (BC) is among the most common female specific cancer types and the second most common cancer in humans after lung cancer [1,2]. Survival rates of BC patients increased in the last years, especially in countries with early diagnostic regimens [3]. 

There are five stages of BC: (i) benign, premalignant stage; (ii) atypical ductal hyperplasia; (ii) preinvasive stage of ductal carcinoma in situ; (vi) metastatic carcinoma; and (v) advanced stage [4]. Known genetic changes in BC include acquired but also inherited changes in oncogenes, tumor suppressor genes and/or genes responsible for genomic stability [5,6]. Nowadays, such changes can be used as biomarkers for BC progression [7]. Moreover, BC can also be grouped according to immunohistochemical markers (ICM), like (i) presence or absence of receptors like those for estrogen (ER), progesterone (PR), human epidermal growth factor receptor-2 (HER-2) or epidermal growth factor receptor (EGFR) on tumor cell surface; (ii) expression of nuclear protein Ki67 as a marker of cell proliferation; and (iii) cytokeratin 5 expression in the plasma of BC cells. These and maybe more ICMs, like TP53 or androgen receptor gene expression, lead to subgrouping of BC in luminal A and B, HER2 positive and negative, and triple-negative or basal like subtypes [8,9,10] (Table 1).

BC subtyping is extremely important for metastasis staging and treatment [7,10,12,13]. In the majority of BC cases, surgical excision of primary tumor is the initial treatment step. Afterwards, radio- and especially chemotherapeutic options are legion; accordingly, various “Clinical Decision Support Systems” are available based on which the potentially most advantageous treatment options for individual patients may be found [14]. Important to mention here is also that gene mutations and specific acquired molecular signatures recently had some impact on the advance of therapeutic targets in breast cancer treatment [15,16].

However, the need for new types of medication with less side effects and being best targeted is still high [3]. Therefore, animal, especially murine, models are regarded as a highly feasible way not only to study biological pathways involved in initiation, progression and metastasis of a tumor like BC but also to establish new targeted medication, like murine tumor cell lines [16,17,18,19]. In spite of the widespread use of such murine tumor cell lines in research, surprisingly, most of them are not characterized molecular cytogenomically [20]. 

Fluorescence in situ hybridization (FISH) is considered the most practicable technique to detect gross genetic alteration in cancer [2]. Thus, in this study, it was taken advantage from combining multicolor-FISH using whole chromosome painting (wcp) probes, FISH banding [21], i.e., murine multicolor banding (mcb), and array-comparative genomic hybridization (aCGH) to do a first cytogenomic characterization of the three murine BC cell lines C-127I, EMT6/P and TA3 Hauschka, which is more than timely, as these cell lines were already established in 1978 [22], in 1986 [23] and in 1953 [24], respectively. Cell lines C-127I and EMT6/P were induced in mice by Harvey virus [22] and anthracycline treatment [23], respectively, while TA3 Hauschka was derived from tumorigenic ascites of a natural murine BC [24]. 

As previously done in comparable studies in murine tumor cell lines, a successful in silico translation from murine to human genome determined the corresponding homologous genetic alteration in human BC and enabled a classification as murine late stage, premalignant stage and benign BC-models.

## 2. Results

### 2.1. FISH Results

#### 2.1.1. C-127I

Murine BC cell line C-127I presented as a pentaploid but was basically genetically relatively instable and was rearranged with more than 10 derivative chromosomes. Consequently, this cell line can be divided to four clones, clone 1 being the ancestor clone and clones 2, 3 and 4 being derivatives of that. 

Clone 1 was present in 20% of the cells and can be described as 94~96,XXX,-2,der(4)t(4;10)(C4;C1),der(4)t(4;10)(C4;C1),+der(4)t(4;5)(C4;F),der(6)(pter→A1::G3→E::B3→D:),del(6)(D),+del(6)(A2),−7,+8,−9,dic(11;18)(A1;A1),inv(12)(BE),−12,−12,del(13)(B1),del(14)(D3),−14,+15,−16,+17,−18,−18,+19.

Clone 2, being present in 40% of the cells had the same karyotype as clone 1 with two additional aberrations, i.e., loss of one chromosome 8 and +der(17)t(11;17)(A3;B3) instead of a normal chromosome 17, as shown in Figure 1.

For clone 3, representing 23% of the cells compared to clone 1, one chromosome 12 was lost and one was replaced by a derivative chromosome 12: del(12)(A1.1).

Clone 4 formed the remaining 17% of cells—here, an additional der(5)t(5;13) and a idic(18) were present as a structural aberrations and one chromosome 10 and 18 each were lost.

#### 2.1.2. EMT6/P

This cell line EMT6/P is triploid and can be divided into five main clones which show some chromosomal instability. Clone 2 can possibly be considered the “ancestor” clone.

The largest clone (clone 1) represents 40% of the cells, with the following karyotype (Figure 2): 59~64,X,der(X)t(X;5)(Xpter→XA1::XA6→XF5::5C3→5qter),der(3)(pter→H1::H1→F2:),+der(3)(pter→H1::H1→F2:),der(4)(pter→C5::E2→C5::C5→D2:),der(5)(5A1→5C3::5B1→5C3::15D1→15qter),+6,idic(8)(A1;A1),+8,idic(10)(A1;A1),idic(12)(A1;A1),dup(13)(C3A2),idic(14)(A1;A1),dup(15)(CA2),+17,+19.

In clone 2 (20%), the breakpoints of der(5)t(5;15) were different than in clone 1 — here, they were in 5B1 and 15D1 as der(5)t(5;15)(B1;D1) compared to a der(5)(5A1→5C3::5B1→5C3::15D1→15qter) in clone 1.

Clone 3 (10%) compared to clone 1 had no normal chromosome 12 and instead a second idic(12)(A1A1). 

Clone 4 (7%) had no additional chromosome 3 and a complex reciprocal translocation t(2;3) with a der(3)t(2;3)(3pter→3H1::2?→2?qter) and der(2)t(2;3)(2?pter→2?::3H1→3F2:): the clone could not be found in mcb2 analyses; thus, breakpoints for chromosome 2 could not be determined. 

Clone 5 (23%) showed reciprocal translocations as follows: t(2;11)(C3;D) and t(9;18); moreover, +der (9) t(9;18)(9pter→9?::18A2→18qter) and +der(18)t(9;18)(18pter→18E2::9?→9qter) were present compared to clone 1.

#### 2.1.3. TA3 Hauschka

This cell line TA3 Hauschka had a particularly stable diploid karyotype and can be divided in three clones for which probably clone 1 was an ancestor of clones 2 and 3. 

Kayrotype of clone 1 is shown in Figure 3 and can be described as follows:

41,X,der(X)(XA1→XA5::19C2→19D2::19D1→19qter),t(1;7)(B;F2),t(1;14)(G;E1),inv(3)(E1H4),del(4)(C3D2),t(6;16)(F3;C3),+6,t(12;13)(E;C1),+16,der(19)t(X;19)(C2;A6),del(19)(D2).

Clone 2 (30%) was characterized by an additional reciprocal translocation t(3;11)(G;E1) and a more complex derivative of chromosome 19, der(19)(19pter→19C2::XA6→XF5::8D1→8qter), compared to clone 1. 

Clone 3 (20%) differed from clone 1 by a der(4)t(4;6;16)(A2;F3;C3) instead of del(4)(C3D2).

### 2.2. aCGH Results

The aforementioned FISH studies of the three murine BC cell lines were in line with the aCGH results and are summarized in Figure 4a, Figure 5a and Figure 6a. A translation of those results to the human genome (only imbalances larger than 3.5 megabase pairs were included in the evaluation) identified the corresponding homologous region in the human genome (Figure 4b, Figure 5b and Figure 6b). Genomic details can be found in Appendix A.

### 2.3. Comparison with Literature

The three studied BC cell lines present acquired copy number variations in regions known to harbor oncogenes and tumor suppressor genes, related to human BC [25,26]; as summarized in Table 2, gains of copy numbers were more frequent than losses. In Table 3, the chromosomal breakpoints observed in the three cell lines are compared to the break events known from human BC; here again, the highest rate of breaks being in concordance with human BC is present for the most advanced cell lines C-127I. Specific DNA copy number alterations correlated with the molecular subtype for human BC [27]—a comparison of the three murine BC cell lines is shown in Table 4. As a result, a high correspondence between C-127I and BC subtype HER2+ and basal-like tumors but also luminal B type was visible.

## 3. Discussion

BC has several subtypes (Table 1) due to the heterogeneity and complex pathology [13]. Still, animal model systems play a major role in BC research, especially for testing new treatment protocols [4,13]. Thus, in this study, the three frequently used murine BC cell lines C-127I, EMT6/P and TA3 Hauschka were for the first time characterized on the molecular cytogenomic level. The feasibility of the applied scheme was shown in several previous studies [20,36,37,38,39].

The three cell lines showed differences in the genomic alteration rate regarding ploidy, numerical and structural aberrations, and tumor-associated breakpoints. C-127I presented a very complex karyotype with pentaploidy, EMT6/P just was triploid and had less structural aberrations than C-127I, while TA3-Hauschka was near diploid and had only few imbalances. Thus, this is the first evidence that the three cell lines may demonstrate different subtypes and stages of human BC [40]. However, polyploidization is common in cancer cell lines [41,42], but polyploidization may also be part of malignancy progression and was also observed in connection with drug resistance [42]. Therefore, the pentaploidy in cell line C-127I may indicate that this cell line may be a suited model for the aggressive stage of BC or the *HER2*-enriched subtype which is known as an aggressive subtype and resistant to treatment [11,34]. 

As all three cell lines were established between 34 and 67 years ago [22,23,24], the chromosomal changes may be both original tumor related or acquired during long times of in vitro cultivation [43,44,45]. Only for TA3 Hauschka, the chromosome number at establishment is known, i.e., 41; thus, this cell line was, as other murine tumor cell lines, remarkably stable over times [24].

As highlighted in Table 2, CNVs could be observed in the three studied cell lines for 7, 10 or 17 of 21 regions known to be locations of tumor suppressor and/or oncogenes in human BC. The most aberrant was again C-127I, followed by EMT6/P and TA3 Hauschka. Interestingly, all three cell lines had gains of copy numbers in the region where the murine *erbb2* gene is localized, while gain in the *brca1* region was only evident for C-127I. Also, oncogene *myc*, being amplified in many human tumors, showed gain of copy numbers in C-127I and EMT6/P but not in TA3 Hauschka. Tumor suppressor gene *rb1* was only deleted in C-127I. 

As according to ploidy, karyotype and CNVs, C-127I is the most aberrant cell line, followed by EMT6/P and TA3 Hauschka; the same tendency is emphasized when looking at the number of chromosomal breakpoints and their localizations (Table 3). The murine chromosomal band homologous to human 9p21, which harbors tumor suppressor genes important in retinoblastoma and *p53* pathway regulation, is affected by chromosomal break events in all three studied murine cell lines. The aforementioned pathways are affected in aggressive forms of BC [28,46]. The murine homologous band to human 19p13.1 is involved also in break events in all three studied cell lines, and this region was identified to comprise genes correlated for enhanced BC risk [29,31,47]. Furthermore, mutations in this region are strongly associated only with ER-negative BC subtypes [33]. Moreover, in TA3 Hauschka and C-127I, breakpoints homologous to human band 14q32 could be observed. Notably, there, the gene *DICER1* is located, recently identified as playing a role in cancer predisposition [30,32]. 

Horlings and coworkers correlated in 2010 [27] the frequency for gains and losses of CNVs to BC molecular subtypes. For this study, only C-127I could be clearly assigned and not only to one but to even three BC subtypes (Table 4): HER2+, basal-like tumors, and luminal B type.

Finally, one may speculate about the influence of breakpoints and/or CNVs of regions being involved in presenting ICM, like ER or PR; in none of the three cell lines, there are changes in regions being homologous to gene estrogen receptor 1 (*ESR1*; in human 6q25.1). However, there was a loss in gene estrogen receptor 2 (*ESR2*; in human 14q23.2~23.3) in C-127I. The latter cell line presents also gains in progesterone receptor gene (*PR*; in human 11q22.1). 

In conclusion, the molecular cytogenomic study and in silico translation for the three here studied BC cell lines, C-127I, EMT6/P and TA3 Hauschka, revealed that they can be used as models for human BC at different stages of malignancy. TA3 Hauschka can be best considered as a model for benign BC, EMT6/P may be used to represent the premalignant or early malignant stage of BC, while C-127I can be the model for the advanced BC stage. These insights are important for future application of these cell lines in BC research and their adequate use.

## 4. Materials and Methods 

### 4.1. Cell Lines

The fibroblast-like/adhesively growing cell lines C-127I (Catalogue number CLS 400134) was purchased from CLS Cell Lines Service GmbH (Eppelheim, Germany), while the EMT6/P- (Catalogue number ECACC 96042344) and TA3 Hauschka-cell line (Catalogue number ECACC 85061102) were obtained from European Collection of Authenticated Cell Cultures (Salisbury, UK). They were grown adherently in DMEM medium (C-127I) or in EMEM/EBSS medium (EMT6/P and TA3 Hauschka) according to company instructions. Cells were prepared cytogenetically as previously reported [20]; whole genomic DNA was extracted using the Blood & Cell Culture DNA Midi Kit (Qiagen, Hilden, Germany) [36]. Cell line-derived chromosome-preparations were subjected to molecular cytogenetic analysis and extracted DNA were subjected to aCGH analysis. Cells were harvested shortly before they reached confluency; other cell density tests were not undertaken.

### 4.2. Molecular Cytogenetics

FISH was performed as previously described [36]. In short, murine whole chromosome paints (“SkyPaintTM DNA Kit M-10 for Mouse Chromosomes”, Applied Spectral Imaging, Edingen-Neckarhausen, Germany) were used for multicolor-FISH (mFISH; results not shown), and murine chromosome-specific mcb probe mixes were used for FISH banding [21]. At least 30 metaphases were acquired and analyzed for each probe set using Zeiss Axioplan microscopy (Carl Zeiss Jena, Jena, Germany), equipped with ISIS software (MetaSystems, Altlussheim, Germany). aCGH was done according to standard procedures by “SurePrint G3 Mouse CGH Microarray, 4x180K” (Agilent Technologies, SantaClara, CA, USA) [36].

### 4.3. Data Analysis

The breakpoints and imbalances of the three studied BC cell lines were determined according to aCGH and mcb data and aligned to human homologous regions using Ensembl Browser, as previously described [37]. The obtained data was compared to genetic changes known from human BC according to literature mentioned.

## Figures and Tables

**Figure 1 ijms-21-04716-f001:**
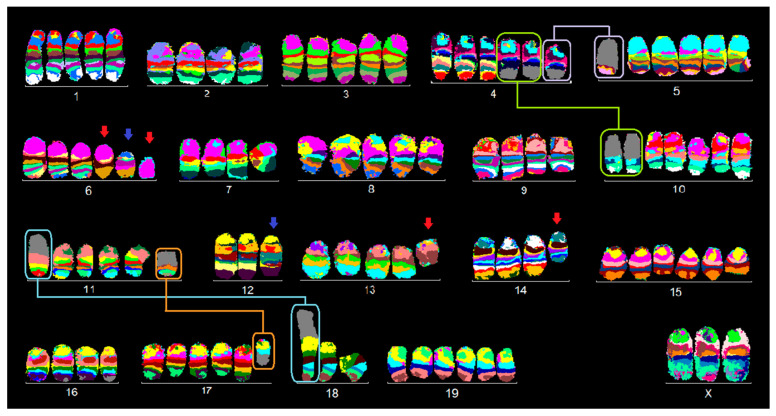
Murine multicolor banding (mcb) was applied on chromosomes of BC cell line C-127I: Typical pseudocolor banding for all 20 different murine chromosomes is shown for clone 2. This figure depicts the summary of 20 chromosome-specific fluorescence in situ hybridization (FISH)-experiments. Four translocations consisting of two different chromosomes each, are highlighted by frames in this summarizing karyogram. Chromosomes with partial deletions are pointed out by red arrows, and chromosomes with inversions are pointed out by blue arrows.

**Figure 2 ijms-21-04716-f002:**
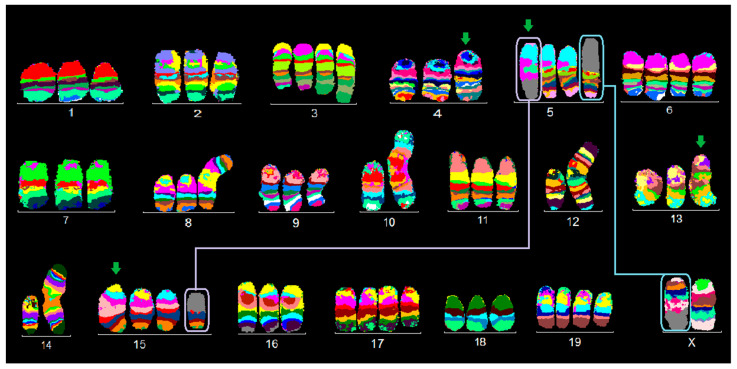
Murine multicolor banding (mcb) was applied on chromosomes of BC cell line EMT6/P: Legend is as for Figure 1. Partial duplications are highlighted by green arrows.

**Figure 3 ijms-21-04716-f003:**
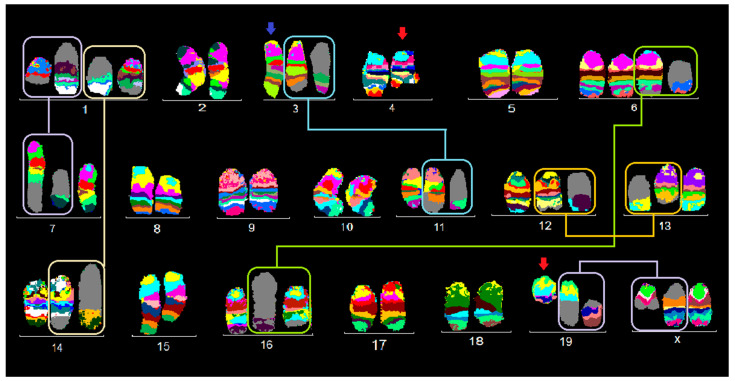
Murine multicolor banding (mcb) was applied on chromosomes of BC cell line TA4 Hauschka: Legend is as for Figure 1.

**Figure 4 ijms-21-04716-f004:**
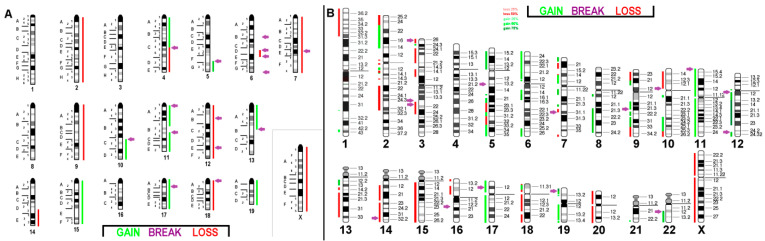
(**A**) Array comparative genomic hybridization (aCGH) results for murine BC cell line C-127I: The copy number alterations with respect to the pentaploid karyotype are given as the color code depicted in the figure with shades of red (losses) and green (gains); purple arrows indicate breakpoints. Breakpoints are indicated according to mcb results. (**B**) Projection of the aCGH results for the cell line onto the human genome showing imbalances as gains and losses of specific chromosomal regions with respect to the original pentaploid chromosome set.

**Figure 5 ijms-21-04716-f005:**
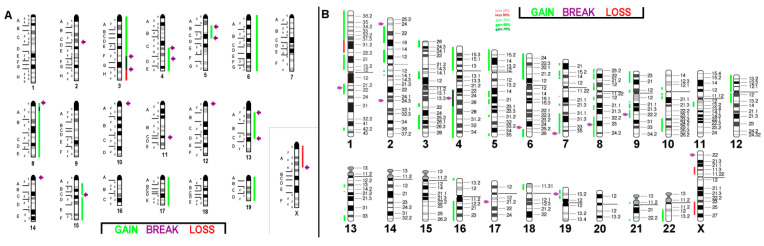
aCGH results for the triploid murine BC cell line EMT6/P (**A**) and its projection onto the human genome (**B**): For more details, see legend of Figure 4.

**Figure 6 ijms-21-04716-f006:**
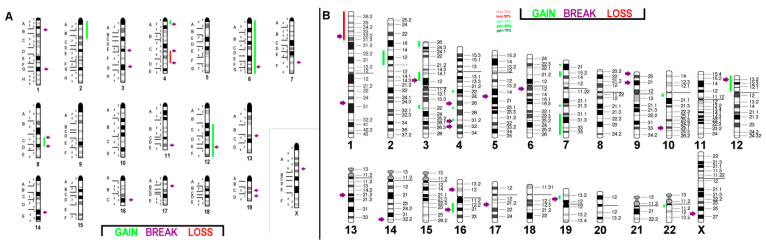
aCGH results for near diploid murine BC cell line TA3 Hauschka (**A**) and its projection onto the human genome (**B**): For more details, see legend of Figure 4.

**Table 1 ijms-21-04716-t001:** Relationship between the molecular breast cancer (BC) subtypes and immunohistochemical markers (ICMs) [11].

Molecular Subtype	ER	PR	HER2
luminal A	+	+	−
luminal B	+	+	−
luminal B	+	+	+
HER-2+	−	−	+
triple negative or basal-like	−	−	−

ER—estrogen receptor; HER-2—human epidermal growth factor receptor 2; PR—progesterone receptor.

**Table 2 ijms-21-04716-t002:** Oncogenes and tumor suppresser genes of importance in BC according to the literature [25,26] and their involvement in gains or loss of copy numbers in the three studied cell lines. Abbreviation: CNV = copy number variant.

Oncogenes and	Gene Loci in Human			
Tumor Suppressor Genes		C-127I	EMT6/P	TA3 Hauschka
*NRAS*	1p22 or p13	gain	gain	no CNV
*MSH2*	2p22	gain	gain	no CNV
*RAF1*	3p25	gain	gain	gain
*RARβ2*	3p24	no CNV	no CNV	no CNV
*MLH1*	3p21	loss	no CNV	no CNV
*APC*	5q21	gain	gain	no CNV
*MYB*	6q22-q23	gain	no CNV	no CNV
*IGFII-R*	6q26	gain	gain	no CNV
*MYC*	8q24	gain	gain	no CNV
*CDKN2A (p16INK4)*	9p21	loss	gain	loss
*PTEN 10q23*	10q23	gain	gain	no CNV
*HRAS*	11p15.5	loss	no CNV	no CNV
*ATM*	11q22	gain	no CNV	no CNV
*CCND1*	11q13	gain	gain	no CNV
*INT2*	11q13	loss	gain	no CNV
*CDKN1B (p27kip1)*	12p13	no CNV	gain	gain
*KRAS2*	12p12.1	no CNV	gain	gain
*BRCA2*	13q12	gain	no CNV	no CNV
*RB1*	13q14	loss	no CNV	no CNV
*CDH1 (E-cadherin)*	16q22	no CNV	no CNV	gain
*TP53 (p53)*	17p13	gain	no CNV	no CNV
*ERBB2*	17q21	gain	gain	gain
*BRCA1*	17q21	gain	no CNV	no CNV
*SERPINB5 (maspin)*	18q21	loss	no CNV	no CNV
*STK11 (LKB1)*	19p13	gain	gain	gain
SUM of concordance in CNVs of potentially affected regions		17/21	10/21	7/21

**Table 3 ijms-21-04716-t003:** Breakpoints in C-127I, EMT6/P and TA3 Hauschka compared to the observed acquired breaks in human BCs according to the literature [6,7,11,25,26,28,29,30,31,32,33,34,35]: Concordances with human breakpoints are highlighted in bold.

Breakpoint Acc. to Human Genome	Human BC	C-127I	EMT6/P	TA3 Hauschka
1p33	+	−	**+**	−
1p13.2	−	−	+	−
1q25.3	+	−	−	**+**
2p23.3	−	**−**	+	**−**
2q31.3	+	−	**+**	−
3p26.1	+	**+**	−	−
3p12.3	+	−	−	**+**
3q14.1	−	+	**−**	**−**
3q21.3	+	**+**	−	−
4p12	−	**−**	+	**−**
4q22.3	+	−	**+**	−
4q26	+	−	−	**+**
4q31.23	+	−	−	**+**
4q32.2	−	**−**	**−**	+
5p14.2	+	−	**+**	−
5q13.2	+	**+**	−	−
5q14.3	+	−	**+**	−
5q15	−	**−**	**−**	+
6q12	+	−	−	**+**
6q25.2	−	**−**	+	**−**
7p14.1	−	**−**	+	**−**
7q31.1	−	+	**−**	**−**
7q36.2	−	**−**	+	**−**
8q23.3	+	−	**+**	−
8q24.22	+	−	**+**	−
9p24.2	+	−	−	**+**
9p21	+	**+**	**+**	**+**
10p11.21	+	**+**	−	−
10q25.1	−	**−**	**−**	+
11p15.5	+	**+**	−	−
12p13.2	−	**−**	**−**	+
12q12.1	+	**+**	−	−
12q24.31	+	**+**	−	−
13q21.2	+	−	−	**+**
14q32	+	**+**	−	**+**
16p12.3	−	**−**	**−**	+
16q13.3	+	**+**	−	−
16q21	−	**−**	**−**	+
17p12	+	**+**	−	−
17q21	+	−	**+**	−
19p13.1	+	**+**	**+**	**+**
20q13.3	+	**+**	−	−
22q12.2	+	**+**	−	−
Xp22.2	−	**−**	+	**−**
Xq23.2	−	**−**	**−**	+
SUM of concordance		27/45	19/45	18/45

**Table 4 ijms-21-04716-t004:** Copy number changes associated with molecular subtypes of human BC, according to [27], with the copy number variants (CNVs) in cell lines C-127I, EMT6/P and TA3 Hauschka: Concordances with human CNVs (in italics) are highlighted in bold. Abbreviations: no CNV = no copy number variants.

DNA Changes in BC Subtypes	Human BC	C-127I	EMT6/P	TA4 Hauschka
**HER2+**				
17q11.1~12	*gain*	**gain**	no CNV	no CNV
17q21.31~23.2	*gain*	**gain**	no CNV	no CNV
***SUM of concordance***		***2/2***	***0/2***	***0/2***
**Basal-like tumors**				
4p15.31	*loss*	no CNV	gain	no CNV
5q12.3~13.2	*loss*	no CNV	no CNV	no CNV
5q33.1	*loss*	**loss**	no CNV	no CNV
6p12.3	*gain*	**gain**	**gain**	no CNV
6p21.1~23	*gain*	**gain**	**gain**	no CNV
8q24.21~24.22	*gain*	**gain**	**gain**	no CNV
10p12.33~14	*gain*	loss	no CNV	no CNV
10q23.33	*loss*	no CNV	no CNV	no CNV
12q13.13~13.3	*loss*	gain	gain	no CNV
15q15.1	*loss*	**loss**	no CNV	no CNV
15q21.1	*loss*	**loss**	no CNV	no CNV
***SUM of concordance***		***6/11***	***3/11***	***0/11***
**luminal A**				
1q21.3~44	*gain*	no CNV	**gain**	no CNV
16p13.12~13.13	*gain*	no CNV	no CNV	no CNV
16q11.2~13	*loss*	no CNV	gain	gain
16q22.1-24.1	*loss*	no CNV	gain	no CNV
***SUM of concordance***		***0/4***	***1/4***	***0/4***
**luminal B**				
1p31.3	*loss*	**loss**	**loss**	**loss**
8p21.2~23.1	*loss*	no CNV	gain	no CNV
17q23.2	*gain*	**gain**	no CNV	no CNV
***SUM of concordance***		***2/3***	***1/3***	***1/3***

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
