# Peer review of "Molecular Cytogenomic Characterization of the Murine Breast Cancer Cell Lines C-127I, EMT6/P and TA3 Hauschka"

_ijms, 2020, doi:10.3390/ijms21134716_

Round 1

Reviewer 1 Report

In the manuscript entitled "Molecular cytogenomic characterization of the murine breast cancer cell lines...." (ID: ijms-853401)" the authors show that three different BC cell lines have different karyotypes, ploidy and chromosomal rearrangements that could be relevant or typical for early or advanced tumor stages.

1. In the Introduction, the authors describe the BC subtypes and the importance for treatment, actually to personalize BC treatment. Several reports about the mutation signatures in many kinds of cancer including BC have been published; I think these studies need to be mentioned in the introduction because they are relevant for therapy and cancer progression, too. Then, the authors have to focus a little bit more their message and establish what new insights the cytogenomic analysis they propose could add to tumor biology or treatment. Importantly, I am not sure that murine tumor cell lines are "animal models". In my opinion animal models are: Mice, Zebrafish or Drosophila; authors should explain that. Are these cell lines directly derived from tumors? such as HeLa or others? 

2. In the Results, the authors make a list of the chromosomal rearrangements and so on. It is not clear why they subcloned the cell lines; in the first sentence of the results (line 73-75), they say "consequently,  this cell line can be divided in four main clones"; I think more explanation is needed also for the other two cell lines.

3. Figure 1-3 are good and legends are quite explicative, However, I would add a title at the figure ex: "chromosomal translocations and rearrangements (deletions, inversions or duplications!?) in ... cell line". Figures 4-6 ( a and b) needed to be described in the text and in the legend. Are those CNV and breakpoint, gain ..loss? These results need to be explained in depth. At line 152, specific DNA copy alteration correlated with the molecular subtype  for human BC and so on; did the authors do some statistic evaluation on that to test the correlation? what does "evident" mean?

Discussion is fine.

Mamd M are described clearly and the references appropriate.

Author Response

Thanks to the reviewer for the suggestions - we did as follows for the points below:

1. A sentence about the mutation signatures in BC have been included and Refs 14 and 15.

- For "Then, the authors have to focus a little bit more their message and establish what new insights the cytogenomic analysis they propose could add to tumor biology or treatment." we added the follwing sentence at the end of abstract "With this cytogenomic information provided, these cell lines now can be applied really adequately in future research studies." and the sentence "These insights are important for future application of these cell lines in BC research and their adequate use." at the end of the Discussionpart. We do not dare to be more speculative for the poetntial impact of the provided data.

- For: "Importantly, I am not sure that murine tumor cell lines are "animal models". In my opinion animal models are: Mice, Zebrafish or Drosophila; authors should explain that. Are these cell lines directly derived from tumors? such as HeLa or others?"

In Introduction part we incldued how the cell lines were established in one more sentece; also a Google search for "murine tumor cell line" "animal model" gave >31000 hits - so murine tumor cell line are considered as animal models in the literature.

2.  For: In the Results, the authors make a list of the chromosomal rearrangements and so on. It is not clear why they subcloned the cell lines; in the first sentence of the results (line 73-75), they say "consequently, this cell line can be divided in four main clones"; I think more explanation is needed also for the other two cell lines." - Done

3. For "Figure 1-3 are good and legends are quite explicative, However, I would add a title at the figure ex: "chromosomal translocations and rearrangements (deletions, inversions or duplications!?) in ... cell line"." This journal does not ask for figure titles

- For: Figures 4-6 ( a and b) needed to be described in the text and in the legend. Are those CNV and breakpoint, gain ..loss? These results need to be explained in depth." Here we tried to understand what may be missing, but we did not find. As we see all is explained in the legends here. As these are quite standard depictions of aCGH results we see no reason to go into more detail. Also we wrote that genomic details for these sumarizing figures can be found in Appendix as Suppl. Table 1.

- For: "At line 152, specific DNA copy alteration correlated with the molecular subtype for human BC and so on; did the authors do some statistic evaluation on that to test the correlation? what does "evident" mean?"

Statistics is here not possible - too low numbers - we just refer to the results shown in Tabs. 2, 3 and 4. Accordingly we replaced the word evident by visible.

Thanks again for reveiwers input, which definitly helped to improve the paper.

Reviewer 2 Report

The proposed manuscript by Azawi et al. presents the results from a cytogenomic characterization study of murine breast cancer cell lines C-127I, EMT6/P, and TA3 Hauschka. Results from the study demonstrated karyotype rearrangement of all three cell lines, as EMT6/P and TA3 Hauschka cells were considered as a model of pre-malignant or early breast cancer stage, while C-127I cells as a model of late stage of breast cancer development.

Nice and intelligently written manuscript that is easy to follow. I do not see issues with data or data interpretations. My following comments are of minor character.

Specific comments and recommendations:

1.Materials and Methods, Section 4.1.: Please provide company catalog numbers of all cell lines. Information about cell culture plate type and seeding cell density also needs to be provided in the text.

2. Materials and Methods, Section 4.3.: It is always preferable if the authors provide all key experimental and analysis details in the text, instead of referring to previous work. In the case of this manuscript, the authors have enough room to extend the Materials and Methods section considerably.

Author Response

Thanks for kind review

For specific comments and recommendations:

1. Materials and Methods, Section 4.1.: company catalog numbers of all cell lines were already included and are now highlighted by 'Catalogue number'.

- Information about cell culture plate type is included now - all are fibroblast like/ adhesively growing

- for seeding cell density: cells were purchased from the suppliers and grown; they were split and harvested  shortly before they reached conflueny; other cell density tests were not undertaken.This is also written now in 4.1

2. Materials and Methods, Section 4.3.: This point was extensively reported in ref 35 - since then we published several papers on other cell lines and all accepted the cross reference to this paper and also the other reviewer had no problem with that short desciption we kindly ask to leave the description for that point as it is. Thanks.